# Transcriptomic Profiling of Breast Cancer Cells Induced by Tumor-Associated Macrophages Generates a Robust Prognostic Gene Signature

**DOI:** 10.3390/cancers14215364

**Published:** 2022-10-31

**Authors:** Meijun Long, Jiajie Wang, Mei Yang

**Affiliations:** 1Breast Cancer Center, Department of Thyroid and Breast Surgery, The Third Affiliated Hospital of Sun Yat-sen University, Guangzhou 510630, China; 2Hunan Key Laboratory of Tumor Models and Individualized Medicine, The Second Xiangya Hospital, Central South University, Changsha 410011, China; 3Department of Orthopedics, The Second Xiangya Hospital, Central South University, Changsha 410011, China

**Keywords:** macrophage, TAM, breast cancer, gene signature

## Abstract

**Simple Summary:**

Macrophage, as the most abundant non-cancerous cell in tumor and one of the major immune cells, plays a critical role in tumor. On the one hand, it is an essential part of immune function. On the other hand, tumor-associated macrophages (TAMs) promote tumorigenesis in multiple ways. To fully analyze TAMs impact on breast cancer and explore the underlying mechanism, we employed a simplified and indirect cell model to identify the genes regulated by TAMs and developed a prognostic gene signature based on them. The gene signature exhibited good prognostic ability in overall survival, distant metastasis-free survival and disease-free survival. It also related with various clinical and biological features of breast cancer, which further strengthens TAMs’ full dimensional role in breast cancer development, encouraging more therapies targeting TAMs and providing a prognostic gene signature with potential clinical applications.

**Abstract:**

Breast cancer, one of the most prevalent neoplasms in the world, continues attracting worldwide attention. Macrophage, as the most abundant non-malignant cell in tumor, plays critical roles in both immune surveillance and tumorigenesis and has become a cell target of immunotherapy. Among all macrophages, tumor-associated macrophage (TAM) is regarded as the main force to promote tumorigenesis. To get an overall view of its impact on breast cancer, we employed a simplified and indirect coculturing cell model followed by RNA-sequencing to detect cancer cell’s transcriptomic response induced by TAM and a prognostic gene signature was constructed based on it. Evidence from both cell models and clinical samples strengthened TAM’s full-dimensional impact on breast cancer, involved in almost all known signal pathways dysregulated during tumorigenesis from transcription, translation and molecule transport to immune-related pathways. Consequently, the gene signature developed from these genes was tested to be powerful in prognostic prediction and associated with various clinical and biological features of breast cancer. Our study presented a more complete view of TAM’s impact on breast cancer, which strengthened its role as an important therapy target. A 45-gene signature from the TAM-regulated genes was developed and shown potential in clinical application.

## 1. Introduction

After so many years, cancer is still an important public health problem and a leading cause of death around the world, and breast cancer is the most commonly diagnosed neoplasm [1]. Although great advances have been made in the clinical management of breast cancer in the last decade, there is still a need for better, simpler and more convenient ways to evaluate prognosis, which is crucial for disease surveillance and optimizing treatment strategies.

Solid tumor is composed of extracellular matrix and a mixed population of both malignant and non-malignant cell types, such as endothelial cells, fibroblasts, stromal cells and inflammatory cells [2]. On the one hand, malignant cells continue evading the immune system by secreting chemotactic tumor-promoting proteins, for example C-C motif chemokine ligand 2 (CCL2) [3,4,5], which directs inward migration and transformation of leukocyte sub-populations (LSPs) to promote tumor initiation and promotion. Types of transformed LSPs within the tumor microenvironment include, but are not limited to, tumor-associated macrophages (TAMs), tumor-associated neutrophils, T-regulatory cells, myeloid-derived suppressor cells, metastasis-associated macrophages and cancer-associated fibroblasts [6]. The crosstalk between malignant and non-malignant cells in tumor is not unidirectional, and non-malignant cells not only provide a favorable microenvironment for tumor growth, invasion and metastasis but also directly enhance the malignancies of tumor cells, including increased proliferation, motility, invasiveness, drug-resistance and metabolism [7,8,9,10,11,12]. Among all the non-malignant cells in tumor, TAMs are the most abundant inflammatory cells and associated with poor patient outcome in multiple types of cancer [9,13,14].

In breast cancer, more than 90% of TAMs are originated from monocytes in the blood [15], which play a critical role in tumorigenesis. In quantity, it is one of the major components of the tumor microenvironment, ranging between 30% and 50% in malignant tissue [16,17]. Functionally, high infiltration of TAMs has been proven to be associated with poor survival rates of breast cancer [18,19,20].

In the present study, we employed the indirect coculturing model of breast cancer cells and TAMs to identify the transcriptomic response of breast cancer cells to TAMs and got a 45-gene prognostic signature from the differentially expressed (DE) genes by LASSO cox regression analysis, which was further demonstrated to be a novel and robust predictor for risk stratification. The result strengthens TAMs’ critical role in the initiation and development of breast cancer and may contribute to strategizing personalized cancer management.

## 2. Materials and Methods

### 2.1. Cell Culture and Induction

MCF7 and MDA-MB-231 breast cancer cells were obtained from American Type Culture Collection (ATCC, Manassas, VA, USA) and maintained, according to standard protocols. Peripheral blood monocytes/macrophages from healthy donors were isolated using CD14 Microbeads (Miltenyi Biotec 130-090-879), according to the manufacturer’s instructions. Monocyte-derived macrophages (MDMs) were grown in DMEM medium (Gibco) supplemented with 10% fetal bovine serum (FBS) (Gibco, Australia), 50 U/mL penicillin and 50 uL/mL streptomycin (Gibco). For the polarization of TAMs, MDMs were treated for 6 days with 30% culture medium from MDA-MB-231 cells, as described [21]. Both MDMs and TAMs were washed with PBS and cultured for an additional 48 h in fresh DMEM 10%FBS, then the mediums were collected, filtered through a 0.22 mm filter to remove cell debris and named as MDMs CM (conditional medium) and TAMs CM, respectively. For induction of MCF7 cells, the cells were cultured with 30% of either MDMs CM or TAMs CM for 6 days.

### 2.2. RNA Sequencing and Transcriptomic Analysis

MCF7 cells in 60 mm cell culture dish with medium containing either MDMs CM or TAMs CM for 6 days were collected for total RNA extraction. Each group included 3 independent replicates. Total RNA was purified from cells using the standard TRIzol method (Life Technologies, Carlsbad, CA, USA) and the quality was determined by Bioanalyzer 4200 (Agilent, Santa Clara, CA, USA). For RNA sequencing, the libraries for all samples were simultaneously constructed using the VAHTS mRNA-seq v2 Library Prep Kit for Illumina^®^ (Vazyme, Nanjing, China). The libraries were also analyzed by Bioanalyzer 4200 for quality control and sequenced using HiSeq X10 system (Illumina, San Diego, CA, USA) with 150 bp paired-end run.

Collected reads were trimmed by Trimmomatic (version 0.36) [22], and clean reads were aligned by HISAT2 (version 2.2.1.0) [23]. The count values and FPKM values for each gene were calculated by StringTie (version 1.3.4) [24] for statistical analysis. The differentially expressed (DE) transcripts between groups (MDMs CM treated vs. TAMs CM treated) were calculated by Ballgown(version 2.6.0) [25]. To compare the whole transcriptomes, principal component analysis (PCA) was performed with FPKM values by R (version 4.1.3) package ggbiplot (version 0.55) and ggplot2 (version 3.3.6) and heatmap of DE genes was produced by R package pheatmap (version 1.0.12).

### 2.3. Data Acquisition of Breast Cancer Cohorts

RNA-seq data (STAR-Counts) for breast cancer (TCGA-BRCA and CMI-MBC) (up to 30 April 2022) were downloaded from the GDC Data Portal (https://portal.gdc.cancer.gov/repository/) (accessed on 30 May 2022), which are the largest and most complete RNA-seq datasets available. Related information was downloaded from cBioPortal (https://www.cbioportal.org/study) (accessed on 30 May 2022). Samples of solid tumor were used. To generate prognostic gene signature and validate its ability, the TCGA-BRCA dataset was randomly divided into training cohort and validation cohort with a ratio of about 2:1, and the CMI-MBC dataset was employed as an additional validation cohort.

### 2.4. Generation of A Gene Signature

To construct an optimal prognostic gene signature, least absolute shrinkage and selection operator (LASSO) penalized Cox regression was utilized to select genes from the DE genes in the TPM value of TCGA-BRCA training cohort by R package glmnet (version 4.1-4). The risks core for each sample was calculated by taking the sum of the LASSO regression coefficient for each signature gene multiplied with its corresponding TPM value. Univariate Cox regression analysis was performed to evaluate the independence of each signature gene expression from overall survival (OS) by R package survival (version 3.3-1) and survminer (version 0.4.9).

### 2.5. Evaluation and Validation of the Gene Signature

To assess the predictive performance of the signature, we conducted time-dependent receiver operating characteristic (ROC) analysis and calculated the under-curve area (AUC) using the R package survivalROC (version 1.0.3). Kaplan–Meier analysis and the log-rank test were also employed to estimate and visualize survival distributions by R packages survival and survminer, in which the patients were dichotomized into high- and low-risk score groups based on the risk scores by cut-off value, determined as the maximum difference between true positive rate and false positive rate in the ROC curve analysis.

### 2.6. GO Enrichment Analysis and Gene Set Enrichment Analysis (GSEA)

To investigate alterations in the molecular signaling pathways, either GO enrichment analysis or GSEA analysis was performed with R package clusterProfiler (version 4.2.2). For the limited number of DE genes in the MCF7 transcriptomes, GO enrichment analysis was performed with the upregulated and downregulated DE transcripts separately. Additionally, with the higher number of DE transcripts in TCGA-BRCA cohort between the high- and low-risk score groups, GSEA analysis was performed to get a more detailed view of the difference in the signaling pathways between transcriptomes.

### 2.7. Drug Sensitivity Prediction

The R package oncoPredit (version 0.2) was employed to predict samples sensitivity to different drugs, based on their RNAseq gene expression data.

### 2.8. Statical Analysis

We used R for data analysis. Unless otherwise stated, the difference between 2 groups was analyzed by *t*-test, Wilcoxon test, or chi square test with R packages ggplot2, according to the data type. For linear regression, R packages tidyverse (version 1.3.2) and ggpubr (version 0.4.0) were used. The type of test and its *p* value were shown. *p* < 0.05 was considered significant.

## 3. Results

### 3.1. The Effects of TAMs on Breast Cancer Cell Transcriptome

To fully explore TAMs’ influence on breast cancer cells, we employed a simplified and indirect cell model, in which the most frequently used breast cancer cell line MCF7 were induced by medium from either MDMs or TAMs to compare differences in transcriptome. To confirm the establishment of the model, we measured and compared cells migration (Appendix A), proliferation (Appendix A) and colony formation (Appendix A) ability. Consistent with reported stimulation of proliferation and motility by TAMs, the MCF7 cells induced by TAMs CM also demonstrated statistically increased motility, proliferation rate and colony formation ability.

In correspondence, the MCF7 transcriptomes responded to different culturing mediums and were completely separated between groups in the PCA plot (Figure 1A). Subsequently, differential expression analysis between MDMs group and TAMs group was performed to find the detailed DE transcripts. In total, 3792 transcripts were observed to be differentially expressed between groups (Figure 1B), including 1536 up-regulated and 2256 down-regulated transcripts in the TAMs group (Appendix A). The functional GO enrichment analysis showed that the immune-related biological processes, especially the negative regulation of type I interferon-mediated signaling pathway, were enriched with up-regulated genes, which is consistent with our TAMs-induced cell model (Figure 1C). Meanwhile, the down-regulated genes were mainly involved in DNA and RNA binding, which subsequently regulated various cellular components, including nuclear envelope, ribosome and spliceosome; endosomal and lysosomal membrane; and growth cone, focal adhesion and cell-substrate junction. Correspondingly, biological processes involving RNA transcription, splicing, transport and translation, as well as protein transport and localization, nuclear division and cell cycle were dysregulated (Figure 1D).

### 3.2. Establishment of the Prognostic Gene Signature from the TAMs-Regulated Genes

To select optimal predictive genes from the DE genes above, we performed LASSO Cox regression with non-zero Cox regression coefficients and 10-fold cross-validation in the random-selected training cohort of TCGA-BRCA dataset, and 45 genes were screened out, including both coding and non-coding genes (Figure 2A,B). The detailed information and coefficients of these 45 genes were shown in Table 1, and the association between their expression and the OS were analyzed by univariate COX regression analysis (Figure 2C). The result indicated that not all but most of the signature genes’ expression was statistically correlated with the OS survival.

### 3.3. Evaluation and Validation of the Gene Signature in Breast Cancer Cohorts

To evaluate the gene signature’s ability in predicting survival, risk scores were calculated by the sum of the expression of each gene multiplied with its coefficient from the LASSO Cox regression. Its OS predictive ability was presented in the TCGA-BRCA training (Figure 3A,B) and validation dataset (Figure 3C,D) separately; its distant metastasis-free survival predicative ability was evaluated in the MBC cohort (Figure 3E,F); the whole TCGA-BRCA cohort was also employed to exhibit its disease-free survival predictive ability (Figure 3G,H). The ROC curves from the TCGA-BRCA training dataset and validation dataset revealed that the risk score exhibited a more stable predicative ability in shorter time points, and the AUC at 3 years was above 0.8 in both datasets. At the time point with the largest AUC, Kaplan–Meier analysis was performed, and the results showed that the OS rate between high- and low-risk score groups were statistically different in both datasets as well (*p* < 0.0001) (Figure 3B,D). Compared with its robust OS predicative ability, the risk score’s ability in predicting the distant metastasis-free survival and the disease-free survival was weaker (Figure 3E,G), but patients dichotomized by the risk score still exhibited statistical difference (*p* < 0.0001) in the Kaplan–Meier analysis (Figure 3F,H).

### 3.4. Association of the Risk Score with Molecular Signaling Pathways

To explore the detailed molecular signaling pathways underlying the predicative ability of the gene signature, we collected samples in TCGA-BRCA cohort with the 50 highest and lowest risk scores separately and compared their transcriptomes by DEseq2 (Figure 4A, Appendix A). It appeared that in all 60,660 transcripts detected, 16,532 transcripts showed statistical differences between groups (*p* < 0.05). Additionally, like the TAMs-induced transcriptomic response in MCF7 cell, more transcripts were decreased in high-risk score group. With so many DE transcripts, the involved molecular signaling pathways were evaluated by GSEA GO analysis (Figure 4B–D, Appendix A). We only presented the top 10 most significant pathways involved in molecular function, cellular component and biological process separately here. It seemed that the DE transcripts were mainly involved in RNA transcription and translation, while immune-related molecular function, including antigen binding and immunoglobulin receptor binding, were most significantly dysregulated (Figure 4B). Similar alterations were observed in pathways involved in cellular components and biological processes; down-regulated DE transcripts were enriched in immune-related pathways and DNA- and RNA- binding related pathways were enriched with up-regulated DE transcripts. This implied inhibited immune function and activated transcription and translation function in the high-risk score group, which were consistent with the positive correlation between the risk score and OS.

### 3.5. Association of the Risk Score with Clinical Features

The clinical features between groups in TCGA-BRCA divided by risk score were compared. Almost all the available clinical features, including T (Figure 5A, *p* = 9.64 × 10^−3^), N (Figure 5B, *p* = 0.03) and stage (Figure 5D, *p* = 0.02) status defined by AJCC pathologic system, as well as PR status by IHC (Figure 5F, *p* = 0.02) and HER2 status by IHC (Figure 5G, *p* = 5.55 × 10^−5^), were statistically different between the two groups, except M status defined by AJCC pathologic system (Figure 5C) and ER status by IHC (Figure 5E). The results also revealed that, at least with current cutoff, the risk score was not a good index to identify the tricky triple-negative breast cancer, which exhibited no statistical difference in ER status by IHC (Figure 5E) between groups, more PR negative patients (Figure 5F, *p* = 0.02) and less HER2 negative (Figure 5G, *p* = 5.55 × 10^−5^) patients in high-risk score groups. Similar and more statistically significant results were observed in the MBC cohort (Appendix A).

### 3.6. Association of the Risk Score with Biological Features

The biological features between high-and low-risk score groups were also compared. For genome instability, all three available indexes, including fraction genome altered (Figure 6A, *p* = 1.5 × 10^−11^), mutation count (Figure 6B, *p* = 0.001) and tumor mutation burden of nonsynonymous mutations (TMB) (Figure 6C, *p* = 0.0049), were statistically higher in high-risk score groups, which corresponded with its worse OS. The MBC cohort also exhibited similar results (Appendix A).

To evaluate the association between the risk score and immune infiltration, we employed the estimated data from the transcriptome data in the place of lacking immune infiltration data. The estimated data of TCGA-BRCA cohort by six algorithms instead of just one were downloaded from TIMER2.0 [26] (http://timer.comp-genomics.org) (accessed on 8 June 2022) to minimize deviation. The differently infiltrated cells between high- and low-risk score groups were shown (Figure 6D, Appendix A). It seemed that the consistency between data by different algorithms was not very good. However, when we focused on the effective cytotoxic immune cells, NK cells and CD8+ T cells, as well as humoral immunity essential B cell, most algorithms estimated less than these cells in high-risk score groups and no statistical difference was estimated by the rest of the algorithms, which again was consistent with its worse OS.

As far as drug resistance was concerned, the drug IC50 of samples in TCGA-BRCA cohort was also estimated based on the transcriptome data and its association with the risk score was evaluated by linear regression (Appendix A). It seemed that the drug sensitivity was affected by the risk score and among 565 drugs, 232 drugs sensitivity was statistically correlated with the risk score, but the correlation was rather small (R^2^ < 0.1). Noticeably, among all CDK4/6 inhibitors analyzed, which are newly approved as first line drugs in breast cancer in recent years, the IC50 of Palbociclib (Figure 7A,B), and Ribociclib (Figure 7C) were positively correlated with the risk score. Additionally, Palbociclib was analyzed twice with similar results.

In conclusion, a novel 45-gene based on TAMs-induced MCF7 transcriptome was developed and validated to have a powerful performance in predicting OS in breast cancer patients. In addition, the model almost completely recovered all clinical and biological features of breast cancer, which showed good potential in clinical application and once again strengthened TAMs full dimensional effects in promoting breast cancer tumorigenesis.

## 4. Discussion

Macrophage, as the most prevalent noncancerous cells in breast cancer, has attracted great attention and contributed to immunotherapy, the breakthrough in cancer treatment in the past decade [27,28,29,30]. The interplay between cancerous cells and macrophage is complicated [31]. Our research provided extra proof and an overall view of the protumor function of TAMs in breast cancer. It demonstrated that the TAMs-induced transcriptomic alterations in breast cancer cells were evident and full dimensional, resulting in the dysregulation of transcription, translation, transport, cell cycle and inhibition of immune response (Figure 1). Almost all confirmed pathways involved in carcinogenesis were somehow related with these pathways. Correspondingly, the DE genes exhibited powerful performances in predicting the patients OS. Considering the cost and effectiveness, a 45-gene signature, including both coding and non-coding genes, was developed from the DE genes, which exhibited good ability in predicting 3- to 5-year OS (Figure 2 and Figure 3). By comparing the transcriptomes of breast cancer samples with different risks cores calculated from the 45-gene signature, more signal pathways were detected to be differentially regulated between the high- and low-risk score groups, but basically these alterations were similar to those in TAMs-induced MCF7 transcriptomes with dysregulated transcription, translation, transport and inhibited immune response (Figure 4). Additionally, we evaluated the relationship between the risk score and various clinical and biological features in breast cancer. It seemed that the majority of available clinical features (Figure 5) and biological features (Figure 6 and Figure 7) were statistically associated with the risk score. Additionally, the widely recognized adverse factors, including higher T, N status and stage, defined by AJCC pathologic system, and genome instability indexed by either fraction genome altered, mutation count or TMB were enriched in a high-risk score group, which corresponded to its worse OS. These data, in vivo and in vitro, once again confirmed the closed-loop control of signals between cancerous cells and macrophages and reiterated the importance of TAMs during carcinogenesis of breast cancer.

When it comes to the clinical application of the 45-gene signature, two things need to be noticed. Firstly, the model failed to identify the tricky triple-negative breast cancer patients, which are characterized by the loss of hormone receptors and HER2 expression. Since the model was constructed to predict patients OS, the failure was understandable. To identify triple-negative breast cancer patients, adjustment of the model’s coefficients or cutoff value of the risk score may be helpful. Secondly, the samples used here includes both treatment-naïve and variously treated samples. Predicting these patients’ OS reflects the patient’s response to these treatments. We have already demonstrated the correlation of the risk score with the estimated IC50 of a variety of drugs, especially CDK4/6 inhibitors (Figure 7 and Appendix A). Considering the derivation of the model and the negative association of the risk score with immune-related pathways and cytotoxic immune infiltrations, it is also possible that the model could be helpful in predicting patients’ response to immunotherapy. Still, more samples and model optimization are needed for this purpose.

## 5. Conclusions

In conclusion, a novel 45-gene signature based on TAMs-induced MCF7 transcriptome was developed and validated to have robust performance in predicting OS in breast cancer patients. In addition, the model almost associated with all clinical and biological features of breast cancer, which showed good potential in clinical application and once again strengthened TAMs full dimensional effects in promoting breast cancer tumorigenesis.

## Figures and Tables

**Figure 1 cancers-14-05364-f001:**
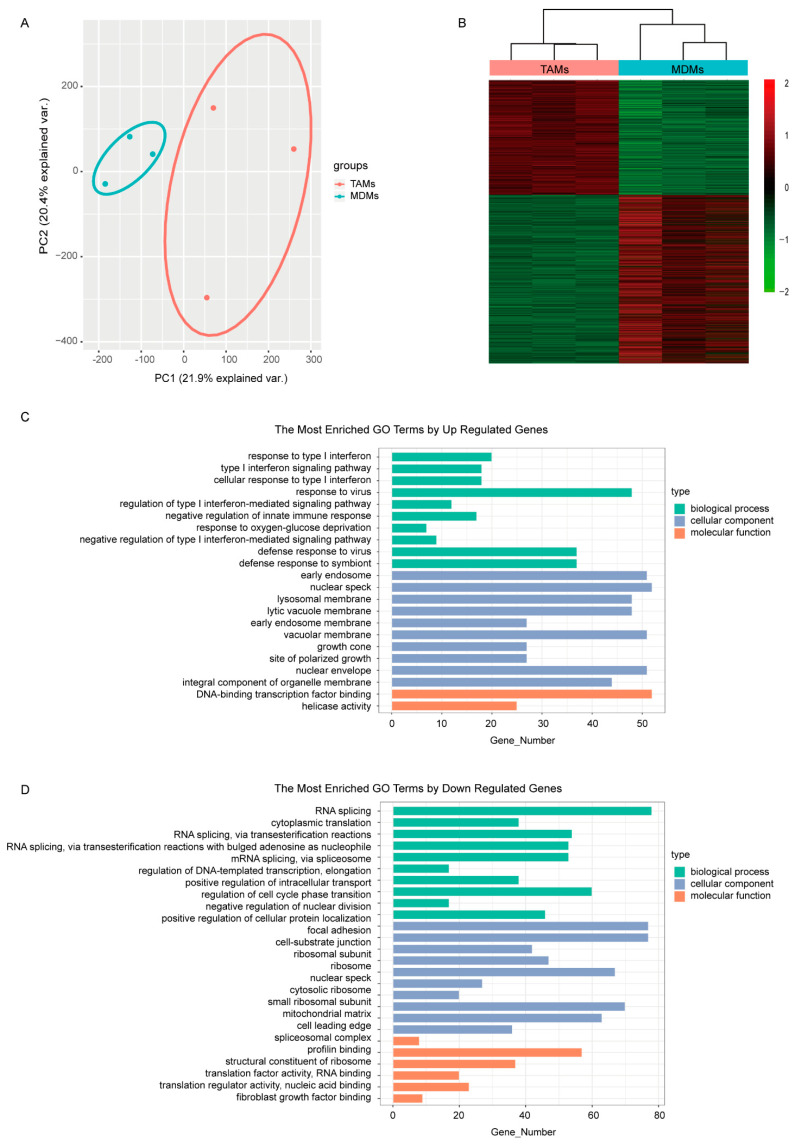
Comparison of the transcriptomes induced by medium from MDMs or TAMs. (**A**) The PCA figure of the MCF7 transcriptomes. (**B**) Heatmap of DEGs in MCF7 transcriptomes induced by either MDMs or TAMs. (**C**,**D**) GO enrichment analysis of the upregulated and downregulated DEGs separately. The pathways shown here were most significantly enriched (*p* < 0.05, q < 0.01).

**Figure 2 cancers-14-05364-f002:**
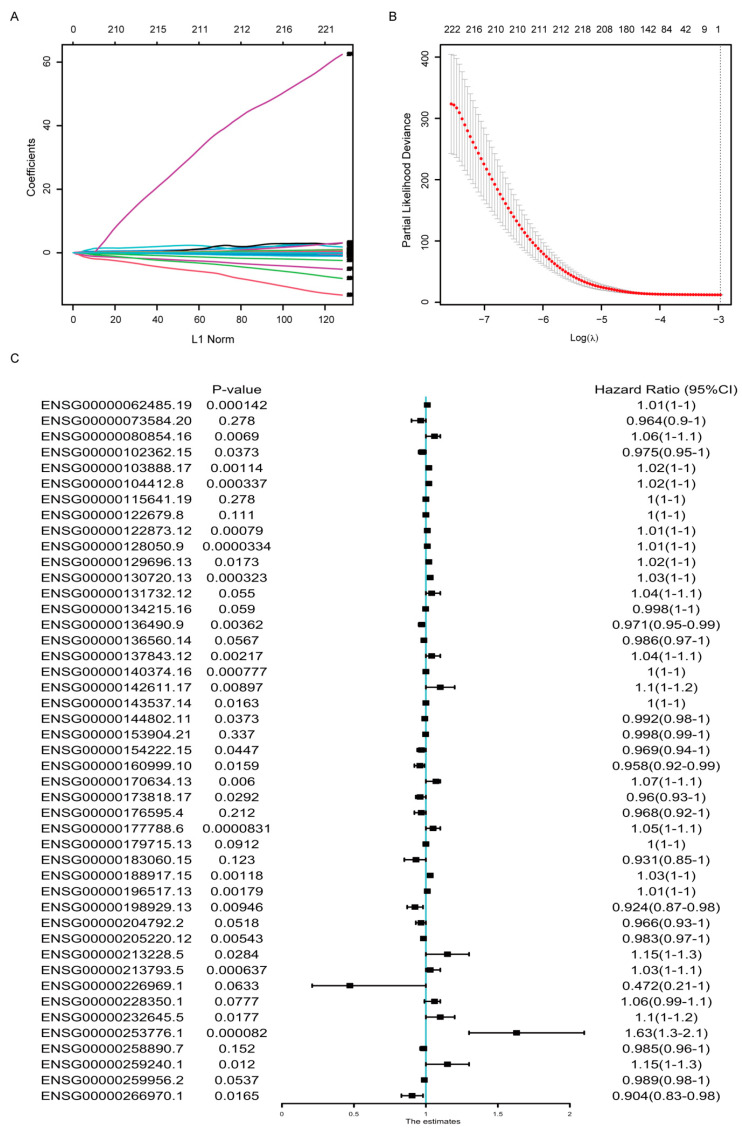
Development of the gene signature based on data in the TCGA-BRCA training cohort. (**A**) LASSO coefficient profiles of the DEGs. (**B**) Ten-time cross-validation for tuning parameter screening in the Lasso–Cox penalized regression model. (**C**) Univariate Cox regression analysis of the relation between the expression of the 45 signature genes and the overall survival in the TCGA-BRCA cohorts. HR, hazard ratio; CI, confidence interval.

**Figure 3 cancers-14-05364-f003:**
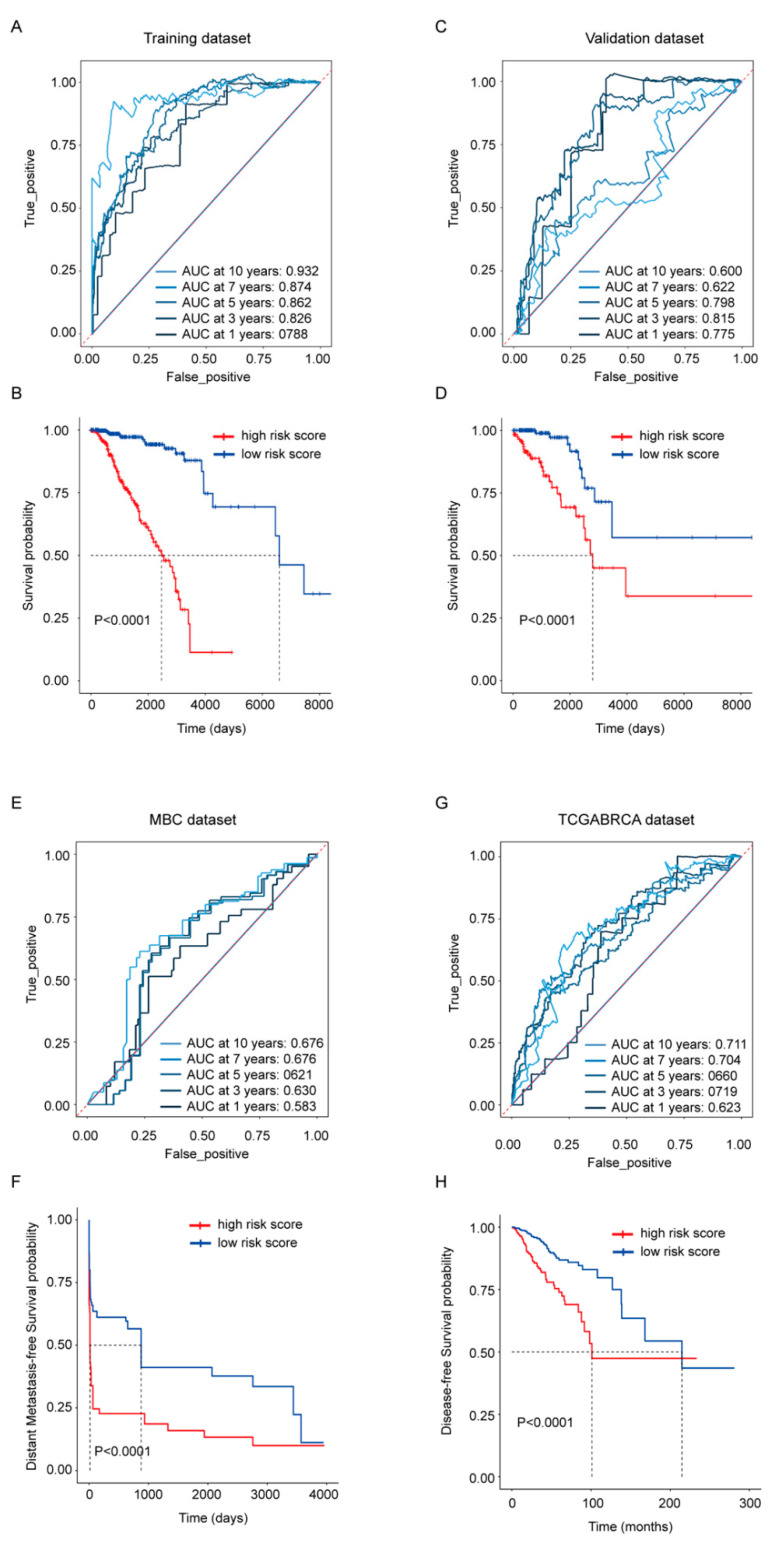
Prognostic performance of the 45-gene signature. (**A**,**C**) Time-dependent ROC curves with AUCs at 1-, 3-, 5-, 7- and 10-year OS based on the gene signature in the TCGA-BRCA training (**A**) and validation (**C**) cohorts separately. (**B**,**D**) Kaplan–Meier plots of OS in subgroups with different risk scores in the training (**B**) and validation (**D**) cohorts separately. (**E**) Time-dependent ROC curves with AUCs at 1-, 3-, 5-, 7- and 10-year distant metastasis-free survival in MBC cohort. (**F**) Kaplan–Meier plots of distant metastasis-free survival in MBC cohort. (**G**) Time-dependent ROC curves with AUCs at 1-, 3-, 5-, 7- and 10-year disease-free survival in whole TCGA-BRCA cohort. (**H**) Kaplan–Meier plots of disease-free survival in whole TCGA-BRCA cohort.

**Figure 4 cancers-14-05364-f004:**
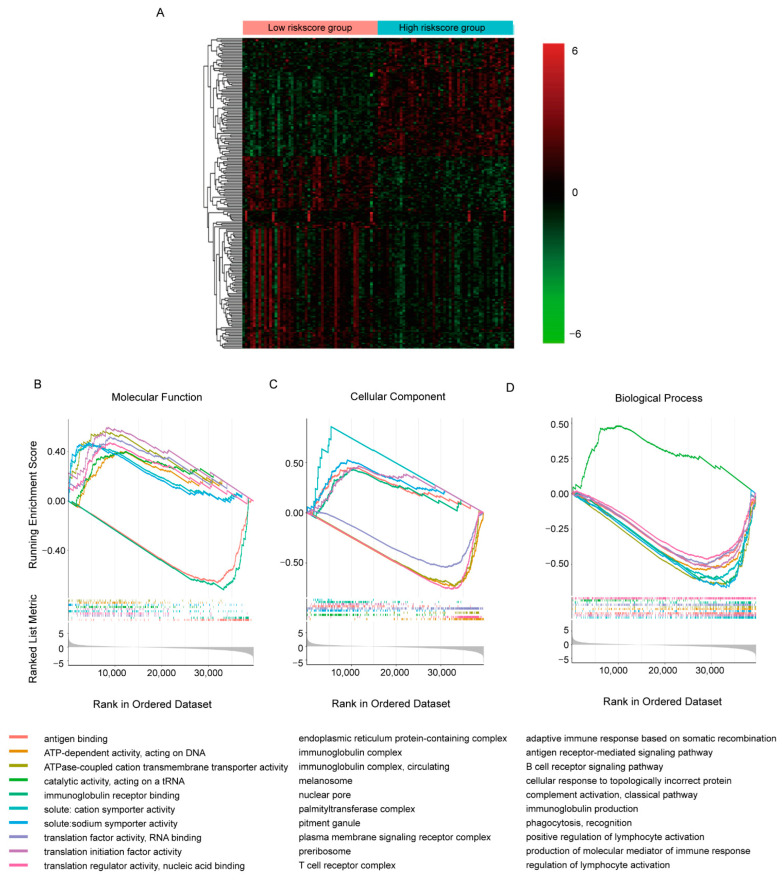
Comparison of the transcriptomes with different risk scores in TCGA-BRCA cohorts. (**A**) Heatmap of top 200 DE transcripts between samples with 50 highest/lowest risk scores. (**B**–**D**) GSEA plots of commonly enriched GO pathways in the TCGA-BRCA cohort. The pathways shown here were top 10 most significantly enriched (*p* < 0.05).

**Figure 5 cancers-14-05364-f005:**
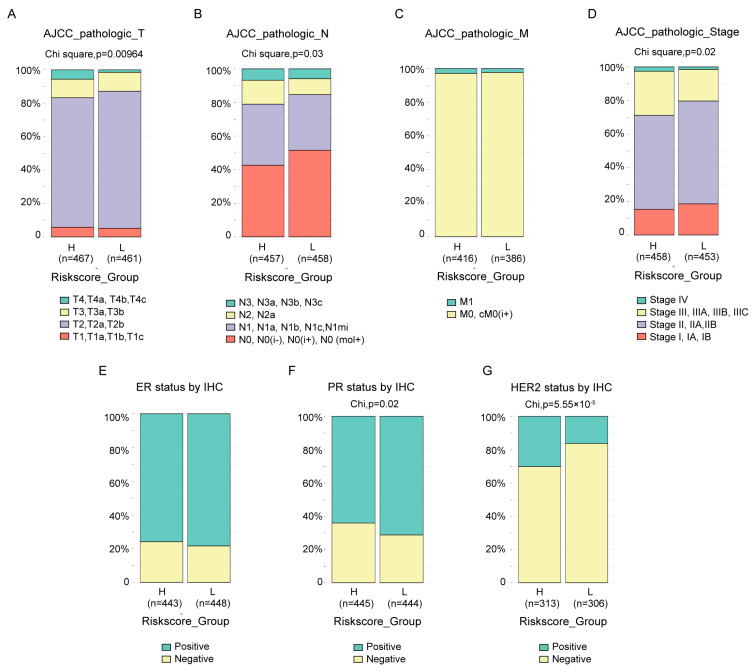
Association between the risk score and clinical features. The difference in the AJCC pathologic T (**A**), N (**B**), M (**C**), stage (**D**) status, ER status by IHC (**E**), PR status by IHC (**F**) and HER2 status by IHC (**G**) between high- and low-risk score groups in the TCGA-BRCA cohort were compared by chi-square test.

**Figure 6 cancers-14-05364-f006:**
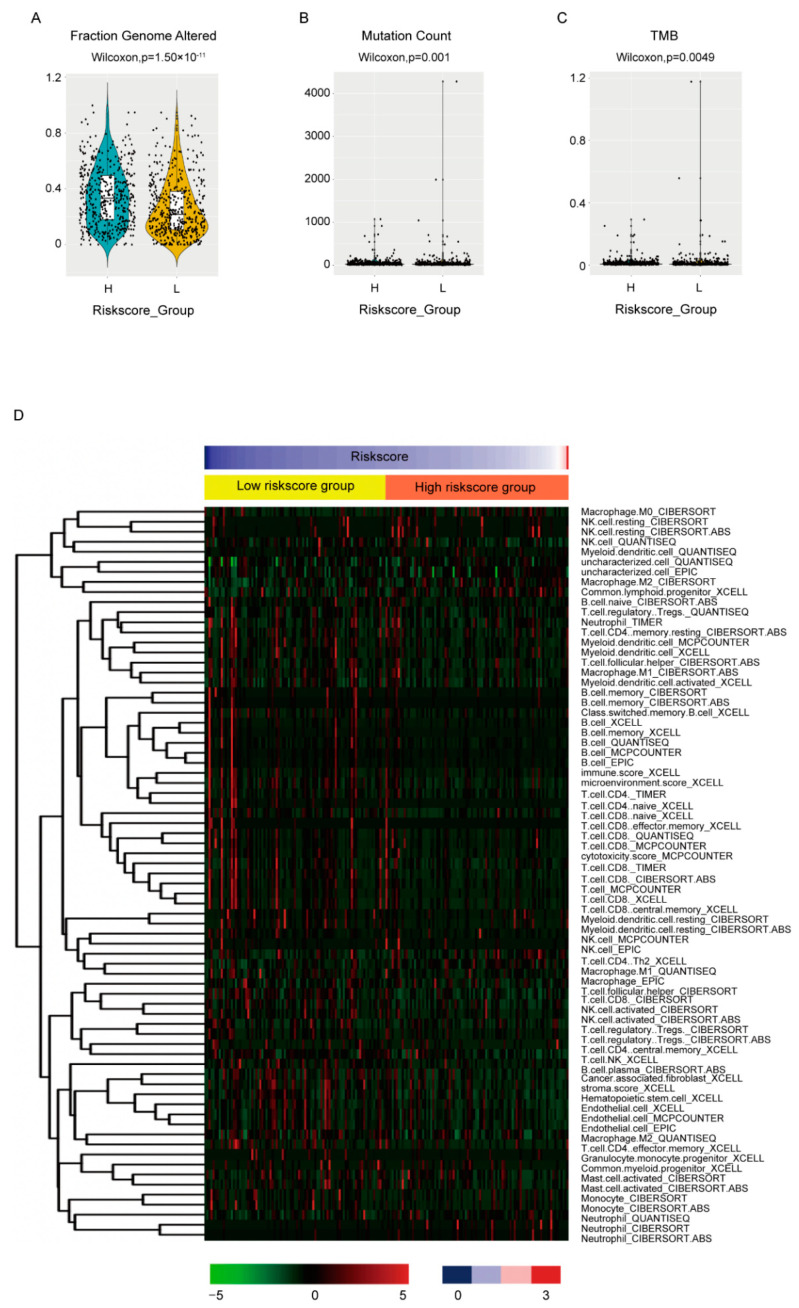
Association between the risk score and biological features. The fraction genome altered (**A**), mutation count (**B**), TMB (**C**) and the estimated immune infiltrations (**D**) were compared between high- and low-risk score groups by Wilcoxon test. Only cell types differentially distributed between groups were shown in (**D**).

**Figure 7 cancers-14-05364-f007:**
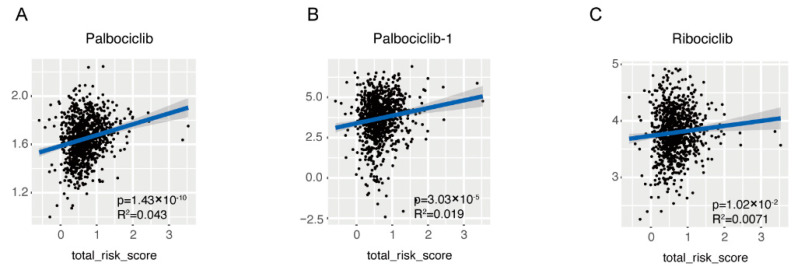
Association of the risk score with the sensitivity of CDK4/6 inhibitor. Linear association between risk score and IC50 of first line CDK4/6 inhibitors was demonstrated in dot plot, including Palbociclib (**A**,**B**) and Ribociclib (**C**).

**Table 1 cancers-14-05364-t001:** Information of genes in the 45-gene signature.

Ensemble. Gene	Gene. Symbol	Description	Coefficient
ENSG00000062485.19	CS	citrate synthase	0.0000856
ENSG00000073584.20	SMARCE1	SWI/SNF related, matrix associated, actin dependent regulator of chromatin, subfamily e, member 1	−0.010460771
ENSG00000080854.16	IGSF9B	immunoglobulin superfamily, member 9B	0.019073486
ENSG00000102362.15	SYTL4	Synaptotagmin-Like Protein 4	−0.002378905
ENSG00000103888.17	CEMIP	cell migration inducing protein, hyaluronan binding	0.002799886
ENSG00000104412.8	EMC2	ER membrane protein complex subunit 2	0.00416519
ENSG00000115641.19	FHL2	four and a half LIM domains 2	0.000287122
ENSG00000122679.8	RAMP3	receptor (G protein-coupled) activity modifying protein 3	0.00014395
ENSG00000122873.12	CISD1	CDGSH iron sulfur domain 1	0.002053503
ENSG00000128050.9	PAICS	phosphoribosylaminoimidazole carboxylase, phosphoribosylaminoimidazole succinocarboxamide synthetase	0.000183216
ENSG00000129696.13	TTI2	TELO2 interacting protein 2	0.000427789
ENSG00000130720.13	FIBCD1	Fibrinogen C Domain Containing 1	0.002554592
ENSG00000131732.12	ZCCHC9	Zinc Finger CCHC-Type Containing 9	0.004186134
ENSG00000134215.16	VAV3	vav 3 guanine nucleotide exchange factor	−0.000553165
ENSG00000136490.9	LIMD2	LIM domain containing 2	−0.002107864
ENSG00000136560.14	TANK	TRAF family member-associated NFKB activator	−0.002189092
ENSG00000137843.12	PAK6	p21 protein (Cdc42/Rac)-activated kinase 6	0.011349395
ENSG00000140374.16	ETFA	electron-transfer-flavoprotein, alpha polypeptide	0.00491458
ENSG00000142611.17	PRDM16	PR domain containing 16	0.030762685
ENSG00000143537.14	ADAM15	ADAM metallopeptidase domain 15	0.001069884
ENSG00000144802.11	NFKBIZ	NFKB Inhibitor Zeta	−0.001081268
ENSG00000153904.21	DDAH1	dimethylarginine dimethylaminohydrolase 1	−0.000327634
ENSG00000154222.15	CC2D1B	Coiled-Coil And C2 Domain Containing 1B	−0.007562522
ENSG00000160999.10	SH2B2	SH2B adaptor protein 2	−0.009621469
ENSG00000170634.13	ACYP2	Acylphosphatase 2	0.018154284
ENSG00000173818.17	ENDOV	Endonuclease V	−0.004457155
ENSG00000176595.4	KBTBD11	kelch repeat and BTB (POZ) domain containing 11	−0.003601533
ENSG00000177788.6	AL162595.1	RAB4A antisense RNA 1	0.013350834
ENSG00000179715.13	PCED1B	PC-Esterase Domain Containing 1B	0.001272099
ENSG00000183060.15	LYSMD4	LysM Domain Containing 4	−0.011804477
ENSG00000188917.15	TRMT2B	tRNA methyltransferase 2 homolog B (S. cerevisiae)	0.002438604
ENSG00000196517.13	SLC6A9	solute carrier family 6 (neurotransmitter transporter, glycine), member 9	0.00489141
ENSG00000198929.13	NOS1AP	nitric oxide synthase 1 (neuronal) adaptor protein	−0.004733383
ENSG00000204792.2	LINC01291	Long Intergenic Non-Protein Coding RNA 1291	−0.001537643
ENSG00000205220.12	PSMB10	proteasome (prosome, macropain) subunit, beta type, 10	−0.001541529
ENSG00000213228.5	RPL12P38	Ribosomal Protein L12 Pseudogene 38	0.032718459
ENSG00000213793.5	ZNF888	Zinc Finger Protein 888	0.008411713
ENSG00000226969.1	AL391845.1	PRKCZ divergent transcript	−0.112396842
ENSG00000228350.1	LINC02585	Long Intergenic Non-Protein Coding RNA 2585	0.001112306
ENSG00000232645.5	LINC01431	Long Intergenic Non-Protein Coding RNA 1431	0.021424453
ENSG00000253776.1	AC099520.2	novel transcript	0.22692143
ENSG00000258890.7	CEP95	Centrosomal Protein 95	−0.000482813
ENSG00000259240.1	MIR4713HG	MIR4713 Host Gene	0.000916309
ENSG00000259956.2	RBM15B	RNA binding motif protein 15B	−0.002009547
ENSG00000266970.1	AC061992.2	SOCS3 divergent transcript	−0.016480999

## Data Availability

The raw RNA-seq data generated in this study are available in GEO under accession number GSE211821. All the remaining data are available within the article, Appendix A, or available from the corresponding author upon reasonable request.

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
