# Peer review of "Transcriptomic Profiling of Breast Cancer Cells Induced by Tumor-Associated Macrophages Generates a Robust Prognostic Gene Signature"

_cancers, 2022, doi:10.3390/cancers14215364_

Round 1

Reviewer 1 Report

In this submitted manuscript, the authors employed bioinformatics analysis approach to identify 45 genes regulated by tumor associated macrophages (TAMs) to promote breast cancer tumorigenesis, providing potential therapeutic targets in clinical application.

Following is something that needs to be improved:

1.     Authors should check the manuscript more carefully, there are some spelling mistakes, such as: in Fig 1A, “MCMs” should be “MDMs”.

2.     The legends of Fig 2 (A) and (B) are misplaced.

3.     The figure legends are scarce and supply very basic information.

4.     Regarding the Fig 7, I just wonder that the association of the risk score with FDA-approved immunotherapy drugs, such as Pembrolizumab and Atezolizumab, or standard chemotherapy drugs, such as Paclitaxel, Doxorubicin and etc. (It is more acceptable, but not necessary to put it in the manuscript).

Author Response

We really appreciate the reviewer’s carefulness and professionalness. According to the reviewer’s suggestions, we have revised our manuscript:

  1. We have carefully checked and corrected the spelling mistakes in Fig1A and some other places in the manuscript.
  2. We have correctly placed the legends of Fig 2A.
  3. We have added more detailed information in figure legends.
  4. We are curious about the risk score’s relationship with the FDA-approved immunotherapy drugs too. And we have searched all the available databases for relevant data. But unfortunately, we could not find any transcriptome data of breast cancer patients receiving immunotherapy with prognostic information. As for the standard chemotherapy drugs, the information of the risk score’s linear association with their IC50 was included in Supplementary Table5: Linear regression analysis result of the risk score and estimated drug IC50 values in TCGA-BRCA cohort.

Reviewer 2 Report

In this study, Long et al. employed a simplified and indirect coculturing cell model followed by RNA-sequencing to detect cancer cell’s transcriptomic response induced by TAM. The authors developed a 45-gene signature from differentially expressed genes in MCF7 cells induced by TAMs, which exhibited powerful prognostic ability in breast cancer. They explored the underlying mechanisms of the gene signature’s prognostic ability through bioinformatics analysis and found that the risk score of the model was associated with multiple clinical and biological features of breast cancers, which reflected the overall effect of TAMs during breast carcinogenesis. Overall, this article evaluates the effect of TAMs during tumorigenesis with real world data from a new point of view, which is interesting and has potential clinical application value. This study provides a potential prognostic 45-gene signature for clinical application. However, there are the following concerns for this manuscript.:

1.     The authors may need to rethink about the title of the manuscript to make clear that the research is about breast cancer.

2.     The authors only validated the gene signature’s prognostic ability in a subset of TCGA-BRCA cohort and an additional MBP cohort. The number of the validation datasets seemed to be small.

3.     The authors took advantage of a simplified and indirect co culturing cell model, which maybe artificial. It would be better to use immunofluorescence to detect the absolute count of TAMs in tissue sections from clinical breast cancer patients and analyze the relevance between the TAMs and 45-gene score. It is also feasible if the authors can find another way to analyze the relationship between TAMs and 45-gene score reversibly.

4.     It is suggested that the authors should detect the dynamic change of the MDMs and TAMs markers by RT-PCR or flow cytometry or another way to ensure inducing MDM or TAMs successfully.

5.     It will be better if more clinical samples can be detected to further verify the prognostic value of the TAMs-associated 45-gene signature, including detection of TAMs, 45-gene score and relative follow-up records, but it doesn’t have to.

Author Response

We really appreciate the reviewer’s thoughtful and professional suggestions. And we have tried our best to revise our manuscript accordingly:

  1. We have revised the title of our manuscript to make it more informatic.
  2. The 45 signature genes include both coding and non-coding genes, which made us difficult to find all the relevant genes in gene chip data and limit our research in RNA-seq data. To make it more difficult, we need relevant survival information to evaluate our gene signature’s prognostic ability. We have searched all the available databases for appropriate dataset, but only TCGA-BRCA and MBC cohorts include all the necessary information. So basically, there is nothing we can do about it right now.
  3. It is a very instructive suggestion. We have considered similar ideas too. The problem is: TAMs, as a subset of highly heterogeneous macrophages with M1 and M2 polarization representing the extremes on each opposing end, are still lacking unique and sensitive molecular markers to precisely identify them in the tissue sections. And the commonly used marker CD206 is not unique for TAMs and only expressed in most but not all of TAMs. Therefore, it is impossible for us to detect the absolute count of TAMs as suggested.
  4. We agree with the reviewer that evidence should be provided to prove the induction success. Therefore, we functionally evaluated TAMs effect on MCF7 cells (supplementary figure1), which was later confirmed in the RNA-seq data of MCF7 cells. We think that the evidence is sufficient. After all, we focused on TAMs’function instead of TAMs in this paper.
  5. Very thankful for the thoughtful suggestion. We plan to do it too, if condition permitted. We are particularly interested to test the gene signature in patients receiving immunotherapy. But it will take some time for us to collect samples and relevant information. Still, we will try.